# Disclosure in Online vs. Face-to-Face Occupational Health Screenings: A Cross-Sectional Study in Belgian Hospital Employees

**DOI:** 10.3390/ijerph18041460

**Published:** 2021-02-04

**Authors:** Jonas Stefaan Steel, Lode Godderis, Jeroen Luyten

**Affiliations:** 1Leuven Institute for Healthcare Policy, KU Leuven, 3000 Leuven, Belgium; jeroen.luyten@kuleuven.be; 2Environment and Health, Department of Public Health and Primary Care, KU Leuven, 3000 Leuven, Belgium; lode.godderis@kuleuven.be; 3IDEWE, External Service for Prevention and Protection at Work, Interleuvenlaan 58, 3001 Leuven, Belgium

**Keywords:** health screening, honest reporting, disclosure, deception, occupational physician, survey

## Abstract

Replacing or supplementing face-to-face health screening by occupational physicians with online surveys can be attractive for various reasons. However, the (cost-)effectiveness of both depends on employees’ willingness to disclose occupational health problems. This article investigates whether employees show a different willingness to disclose information in online surveys compared to face-to-face consultations with an occupational physician. Employees from four Flemish hospitals were asked whether they would disclose a range of typical occupational health problems to either surveys or physicians. The results were analyzed through chi-square tests and multilevel ordinary least squares regression. Of the 776 respondents, 26% indicated that they did not always disclose health problems. Respondents were more inclined to disclose mental health problems to a survey than face-to-face to a physician, whereas the opposite was true for medication misuse. Being male, younger, with lower educational attainment or lower trust in physicians, taking medication, or having a lower risk on alcohol abuse increased the likelihood of a person withholding information. We conclude that this study provides indications that online vs. face-to-face health check-ups have different strengths and weaknesses in this respect. These must be considered when evaluating the need to use online surveys (instead of, or together with, face-to-face contacts) for health screening.

## 1. Introduction


*“Keep a watch also on the faults of the patients, which often make them lie about the taking of things prescribed”*
(Hippocrates of Kos, Decorum)

In many countries, e.g., 21 of the 27 EU member states [1], employees are mandated to undergo periodic health screenings by occupational physicians to assess fitness for work and to timely detect potential work-related health impairments and diseases or changes in general health that may have consequences in the workplace (e.g., musculoskeletal problems, substance abuse at work, or mental health risks, such as depression or burnout). In the EU, these routine annual medical examinations can be organized regularly for all workers (14 out of 27 EU countries) or only for some groups, e.g., when there is exposure to specific occupational hazards (12 out of 27 EU countries) [1]. As such, these consultations also contribute substantially to public health, especially given their broad application.

However, a common challenge of such screenings is that their costs are substantial and they are demanding in terms of working hours of an increasingly pressurized group of occupational physicians [2,3,4]. While a face-to-face consultation is still the norm, employers and occupational health services alike are therefore likely to be attracted to digital alternatives, such as screening surveys. These are plausibly less costly, less time-intensive, more flexible, and potentially a more effective way of screening, as occupational physicians can subsequently dedicate more time to employees for whom a health impairment, disease, or reduced functioning at work has been detected. Even if one argues that surveys cannot fully replace face-to-face contact with a physician, a hybrid approach (e.g., alternating surveys with face-to-face contacts, or using a survey to screen for high risks and selectively refer these employees) could have the same effect.

A key assumption underlying this trend is that respondents are equally willing to disclose potential problems to a screening survey as they are during health screenings face-to-face with their occupational physician. However, this equality is far from self-evident. Physicians tend to have what Burgoon calls a ‘truth-bias’: they assume that the information they receive from patients is truthful, complete, and accurate [5,6]. In contrast, from previous research in other medical settings (and even since ancient Greek times), it is known that patients are not always honest in their answers during face-to-face conversations with their physicians [6,7,8]. Patients may fabricate false information, withhold information, exaggerate or embellish, mix truthful and deceptive information, or obfuscate by implying false conclusions or misdirecting attention away from the issue at hand [6]. However, it would be wrong to assume that concealing information is always intentional or morally questionable. It may also happen accidentally because patients respond in a socially desirable way [9], find it hard to remember certain details (e.g., when asked to report medication adherence [10]), or find certain information private or difficult to share.

Surveys, on the other hand, are often used for the anonymity and practical ease they offer [11,12], especially when measuring delicate topics, such as mental health or illegal activities. Several studies have demonstrated that when information is regarded as sensitive (e.g., it addresses a taboo topic, it induces a fear that information might be disclosed to a third party or that the answers are in conflict with the prevalent social norm), anonymous surveys increase self-disclosure [13]. In addition, computerized surveys have been shown to increase self-disclosure of sensitive information when compared to paper-and-pencil questionnaires, possibly because they create an illusion of privacy while being immersed into another virtual world (even when this is merely a belief, rather than actual observation of their answers) [13].

However, it is unclear whether these results are robust when broadening the scope beyond sensitive information (e.g., drugs, alcohol, or other sensitive behavioral information) by also examining the disclosure of health and functioning problems. In addition, transferring these conclusions to an occupational health setting is also problematic, as the problem of honest reporting can come to the fore even more strongly here. The employee could have an incentive to exaggerate conditions in order to gain adjustments to his workload (e.g., feigning sleeping problems to avoid night work), or the opposite: he could underreport his alcohol misuse to ensure he can keep his job in the transport sector. Other reasons to not report health problems could also play a role, such as trust in the occupational physician [14]. While the physician is bound by a duty of professional confidentiality, this is not always known or believed. It is sometimes wrongfully assumed that occupational physicians serve employers more than employees [15], and, as stated above, the belief of privacy can be more important than actual privacy.

Furthermore, while surveys circumvent certain biases, they also raise several questions regarding the accuracy of patient disclosure, as their validity and reliability should be (but is not often) tested extensively. Following the COSMIN taxonomy of measurement properties, questionnaires should ideally have their internal consistency, reliability, measurement error, content and face validity, criterion validity (concurrent and predictive), construct validity (both structural validity, hypotheses-testing, and cross-cultural validity), responsiveness, and interpretability tested in multiple studies [16,17]. Even when care is taken to assess the validity and reliability of a questionnaire, it is not always straightforward to compare the accuracy and completeness of survey responses to a physician who can be more responsive to context and non-verbal communication [14].

In this article, we investigate employees’ self-reported disclosure of responses to the occupational physician and to a survey. We analyze which factors influence disclosure in general and whether this differs between occupational physicians versus surveys. To prevent biases in the survey responses, care was taken to make use of validated and standardized questionnaires. To our knowledge, this is the first study attempting to do so in this setting.

## 2. Materials and Methods

As part of a larger study, a survey was conducted in four Belgian hospitals, between June and October 2019, among all employees eligible for a yearly periodic health screening by an occupational physician in 2019 (safety functions, jobs with heightened vigilance, work that involves physical, biological, or chemical agents or tasks that are an ergonomic or mental burden). The larger study aims to develop a new medical screening tool for periodic medical examinations [18] and to assess the (cost-)effectiveness of occupational health screening of hospital employees when comparing face-to-face consultations by an occupational physician with a screening survey followed by selective follow-up by the physician. The protocol for this study has been published on Clinicaltrials.gov (identifier NCT04684316). We made use of the baseline data from this study before an intervention was carried out. Participation was voluntary, and the introduction mentioned that answers were treated confidentially.

The survey contained two questions on whether the employee intended to disclose a range of potential problems: “Would you honestly disclose a problem with/having concerns about your physical health, mental health, functioning at work, medication misuse, and lifestyle in a conversation with your occupational physician?” and “Would you honestly disclose a problem with/having concerns about your physical health, mental health, functioning at work, medication misuse, and lifestyle in a survey?” Employees could answer “yes” or “no” for each potential problem (physical health, mental health, functioning at work, medication misuse, and lifestyle) in both modes (survey and face-to-face conversation). In the analyses, the categories were summed up (0 for no, 1 for yes), separately for surveys and physicians (yielding a score of 0–5) and jointly for all (yielding a score of 0–10).

Other questions from the survey were socio-demographic characteristics (age, gender, income, nationality, household type, place of work/hospital, occupation), various health state information (including the Nordic Musculoskeletal Questionnaire [19], Nordic Occupational Skin Questionnaire [20], General Health Questionnaire [21], Need For Recovery Scale [22], EQ5D visual analogue scale [23], Medical Research Council (mMRC) questionnaire) [24], and lifestyle (the Alcohol Use Disorders Identification Test (AUDIT-C) [25]) was measured. In addition, parts of the Copenhagen Psychosocial Questionnaire [26] and iMTA (Institute for Medical Technology Assessment) Productivity Cost Questionnaire (iPCQ) [27], and the Trust in Physician Scale [14] adjusted to the occupational health context were included in the survey and used in the analyses. We hypothesized that all these variables can potentially influence reporting behavior, and we therefore tested their relevance in the regressions.

A chi-squared proportions test was used to test whether the proportion of people who would not disclose a problem to the occupational physician in a category (physical health, mental health, functioning, medication, or lifestyle) was significantly different from the proportion of people that would not disclose a problem on the survey on that category. With these tests, we therefore aimed to investigate whether and on which topics employees disclose more to the physician or to a survey. Multilevel ordinary least squares (OLS) regression was then used to assess what might influence the concealing of information in the population against the occupational physician and on the survey. To avoid multicollinearity, the correlations between the independent variables were assessed, the variance inflation factors (VIFs) were calculated, and a Lasso regression was performed to identify the most important predictors. All analyses were performed in R version 4.0.3.

All aspects of this study received ethical approval from the Ethical Committee of UZ/KU Leuven (ref number S62259 and OBC number MP008817). Only anonymized data were obtained by the researchers. Informed consent was obtained from all participants. Due to the sensitive nature of the data, no data can be made available publicly.

## 3. Results

Of the 3150 eligible employees, 869 employees participated (776 fully completed the survey), meaning that, on average, 28% of the eligible population responded (and 25% fully completed the survey). Descriptive statistics of the sample are shown in Table 1. Cut-offs were made in accordance with the questionnaire sources (e.g., burnout risk of the Copenhagen Psychosocial Questionnaire (COPSOQ) or breathlessness in mMRC). Similar to other samples taken from the hospital sector, the majority was female and more highly educated. However, we wish to note that a wide variety of occupational backgrounds was included in the study (e.g., technicians, administrative personnel, caretakers, cleaning staff, nurses, and physicians all took part in the survey) and that hospital personnel are typically prone to a wide range of risks (physical, chemical, psychological, and biological).

Table 2 states the incidence of intention to report in the sample. Depending on the occupational health problem, 6% to 14% indicated that they intend to withhold information, particularly for problems of mental health, lifestyle, or medication misuse. It should also be noted that Table 2 only gives insight in the intended reporting behavior for each unique occupational health problem. When looking across categories, 200 employees (26%) stated that they intended to conceal information at least once from the survey or physician, 145 (19%) stated that they intended to conceal information at least once from the physician, and 110 (14%) stated that they intended to conceal information at least once from a survey.

The chi-squared proportions tests indicated no differences between surveys and physicians for intention to disclose physical health problems, functioning at work or potential lifestyle issues. However, there were differences between surveys and physicians to disclose mental health problems and misuse of medication. Respondents were more likely to disclose mental health problems to a survey than to a physician (*p*-value = 0.003), whereas they were more likely to disclose medication misuse in-person than to a survey (*p*-value = 0.024).

The ordinary least squares (OLS) regression results are presented in Table 3. A first model included gender, age, income, nationality, household status, hospital, education, trust in physician, relationship with the occupational physician, EQ5D visual analogue scale, need for recovery, general mental health, medication use in the last 2 weeks, medication dependence, alcohol use disorders, musculoskeletal health, absenteeism, burnout risk, stress risk, respiratory problems, and skin problems. After analysis of the VIF values and a lasso regression with said variables, income, nationality, household status, need for recovery, musculoskeletal health, absenteeism, burnout risk, stress risk, respiratory problems, and skin problems were removed from the model, rendering the model in Table 3 below. In this final model, all VIF values were below 2, indicating a low risk of collinearity. The results indicate that being male, younger, with lower educational attainment, lower trust in the physician, having taken medication in the last 2 weeks, and taking medication recreationally all increase the intention to conceal information. Finally, a higher risk of alcohol abuse decreases the intention to conceal information overall, where this effect is slightly stronger for reporting to physicians.

## 4. Discussion

In this article, we analyzed the self-reported intention of employees when they are asked to disclose various plausible occupational health problems either to an occupational physician or to a survey. More than one quarter (26%) of the respondents did not always intend to disclose problems, which suggests an unexpectedly high false negative rate in both surveys and in-person contacts and indicates that the effectiveness of health screening can be markedly increased if individuals can be persuaded into more accurate disclosure.

In the analyses, many of the explanatory factors conformed to our expectations, e.g., lower trust in the physician lowers intention to disclose problems. Educational attainment, which is strongly associated with socioeconomic status, was among the main explicatory variables of intention to conceal, indicating the need to pay extra attention to this already disadvantaged group of workers. Tailoring the measurement of occupational health problems to their needs could help safeguard the needs of this vulnerable group [28]. The same holds for young males, who have not yet had the chance to build a trustworthy relationship with their occupational physician. Another relevant finding is that employees with higher medication use or recreational medication use are more prone to hide information from a survey or a physician, while it is exactly those employees who need follow-up the most.

However, on a positive note, the results of the AUDIT-C (alcohol use disorders) questionnaire in the regression do suggest that alcohol misuse increased the employees’ tendency to disclose more. This could be related to the specific Belgian context, as a Collective Labour Agreement (CLA no.00) demands that all private organizations in Belgium must have a policy statement on alcohol and drugs in the workplace and also promotes the development of an appropriate prevention policy [29]. This CLA, together with the fact that OPs are regularly in contact with a significant proportion (at least 70%) of the working population (mostly in a preventive medical setting), might have successfully reduced the stigma of alcohol abuse [30].

The results were largely robust across both surveys and physicians, and there were few (significant) differences between both. However, there are indication surveys would cause less concealment when reporting mental health and more concealment when reporting medication problems (refer Table 2). This might point to a differing value of anonymity (for mental health, see e.g., [11]) and a reluctance to commit information to paper in fear of sanctioning, legal consequences, or even job loss by the employer (for medication misuse) [31].

The substantial degree of concealment can also be seen in the context of a hard-to-eradicate misconception that the occupational physician serves employers more than employees [15] and a fear that complaints will be passed on to the employer without consent by the employee [32]. It could therefore prove beneficial to highlight the anonymity and confidentiality of health complaints to improve their reporting and to continue efforts to clarify the role of the occupational physician. The high false negative rate in both surveys and in-person contacts also emphasizes the importance of diversifying measurement methods. Measuring occupational health problems repeatedly, for instance with both surveys and in-person contact, as well as objective methods where possible (e.g., biometrics, blood tests), or even with hybrid methods [33], can increase the accuracy of incidence estimates and can help to timely detect occupational health complaints. For instance, if it is confirmed that information from surveys on mental health is more accurate, one could use surveys to gather this specific information and subsequently discuss this in face-to-face consultations. While the occupational screening context has unique features (e.g., the incentive to misreport health complaints to gain work adjustments or permissions), this conclusion can also be extended to other medical settings (especially public health screening), as the truth-bias of the physician, concealment by the patient, and accuracy and completeness of survey reports are likely to be similar in these settings.

Limitations of this study were the response rate (25%, which was low but in line with other surveys) and the fact that the study took place only in the hospital sector. However, hospitals also house a diverse population (e.g., technicians, administrative personnel, caretakers, cleaning staff), who all took part in the survey. Another limitation is that we, for our explorative purposes, only investigated intention to disclose problems through one set of questions, relying on self-reports of employees. Further research should corroborate our findings by studying more in-depth the degree and motivation of (intentions toward) disclosing problems and use more diverse populations to extend our conclusions to other public health settings. In addition, future studies could clarify the extent of reporting false positives (reporting a problem while there is none) versus false negatives (not reporting a problem while there is one). Since cross-cultural differences can play a role in disclosure [34] and occupational health practices vary internationally [35], studies in different cultural settings would also contribute to these findings.

## 5. Conclusions

Digital surveys are an attractive alternative to face-to-face consultations in occupational health screening, but an important variable to consider is whether respondents are equally likely to disclose their potential problems using a survey. In this study, more than one quarter (26%) of the respondents indicated that they would conceal information from either a physician or a survey. Although these differences where small in magnitude, respondents were more inclined to disclose mental health problems to a survey than to a physician, while the opposite was true for the misuse of medication: respondents were more inclined to disclose misuse of medication to a physician than to a survey. The high false negative rate in both surveys and in-person contacts is an important and often neglected aspect of the (cost-)effectiveness of both in-person or online screening programs. Investing in a trustworthy patient–physician relationship can help to ensure the reporting of potential problems. This study thus provides indications that online vs. face-to-face health check-ups have different strengths and weaknesses with respect to reporting occupational health problems. These must be considered when evaluating the need to use online surveys (instead of, or together with, face-to-face contacts) for health screening.

## Figures and Tables

**Table 1 ijerph-18-01460-t001:** Summary table of sample (N = 776); numeric variables state mean and standard deviation (SD); categorical variables state count and proportion.

Variable	Levels	Mean/Count	Proportion/Standard Deviation
Gender	Male	143	18.5%
	Female	631	81.4%
	Other	1	0.1%
Age	Numeric [0–66]	45.39	11.2
Income	Income 1500.00–1999.99 monthly	240	37.1%
	Income 2000.00–2499.99 monthly	241	37.2%
	Income 2500.00–2999.99 monthly	76	11.7%
	Income 3000.00–4999.99 monthly	26	4%
	Income > 5000.00 monthly	5	0.8%
	Income not stated	59	9.1%
Nationality	Belgian	763	98.3%
	Migrant inside EU	9	1.2%
	Migrant outside EU	4	0.5%
Education	No degree	7	0.9%
	Primary education	23	3%
	Secondary education	141	18.2%
	Higher education	588	75.8%
	Other education	17	2.2%
Household	Single	89	11.5%
	Single with child(ren)	63	8.1%
	Couple with child(ren)	480	61.9%
	Couple without child(ren)	122	15.7%
	Other	22	2.8%
Hospital	Hospital A	197	25.4%
	Hospital B	266	34.3%
	Hospital C	214	27.6%
	Hospital D	99	12.8%
Occupation	Medical Personnel	394	52.6%
	Paramedics	20	2.7%
	Technicians	56	7.5%
	Administrative Personnel	120	16%
	Management	34	4.5%
	Cleaning Staff	34	4.5%
	Other occupations	91	12.1%
Musculoskeletal (NMQ) ^1^	Complaint last 12 m	701	90.3%
	Complaint last 7 d	558	72.0%
	Functioning complaint	358	46.1%
Skin (NOSQ) ^2^	Skin complaint	155	20.0%
General Mental Health (GHQ) ^3^	Numeric [0–12]	2.07	2.9
Need for Recovery (NFR) ^4^	Numeric [0–11]	3.78	2.8
EQ5D Visual Analogue Scale	Numeric [0–100]	78.95	14.6
Respiratory (mMRC) ^5^	Occasional breathlessness	141	18.2%
	Severe breathlessness	40	5.2%
Alcohol Use Disorders (AUDIT-C) ^6^	Numeric [0–12]	2.03	1.6
Burnout (COPSOQ) ^7^	Moderate risk of burnout (50–74)	286	37.9%
	High risk of burnout (75–99)	98	12.6%
	Severe risk of burnout (100)	3	0.03%
Stress (COPSOQ) ^7^	Numeric [0–12]	4.01	2.5
Absence (iPCQ) ^8^	Absent at least once last 4 weeks	74	0.1%
	Days absent last 4 weeks [0–28]	1.34	5.4
Trust in Physician score	Numeric [0–55]	37.85	6.17
Relationship with the occupational physician (OP)	Known OP for more than 5 years	327	42.2%
	Known OP for 3–4 years	135	17.4%
	Known OP for 1–2 years	165	21.3%
	Known OP less than 1 year	147	19%
Prescribed medication use in the last 2 weeks	Yes	397	51.2%
Medication dependence—used recreationally	Never	718	92.6%
	Monthly or less	44	5.7%
	2–4 times per month	9	1.2%
	2–3 times per week	2	0.3%
	4 or more times per week	2	0.3%

Missing values were removed from this table. ^1^ Nordic Musculoskeletal Questionnaire (NMQ), ^2^ Nordic Occupational Skin Questionnaire (NOSQ), ^3^ General Health Questionnaire (GHQ), ^4^ Need for Recovery Scale (NFR), ^5^ Medical Research Council Dyspnoe questionnaire (mMRC), ^6^ Alcohol Use Disorders Identification Test (AUDIT-C), ^7^ Copenhagen Psychosocial Questionnaire (COPSOQ; burnout questions are based on the Copenhagen Burnout Inventory), ^8^ iPCQ = iMTA (Institute for Medical Technology Assessment) Productivity Cost Questionnaire (iPCQ).

**Table 2 ijerph-18-01460-t002:** Incidence of disclosure to occupational physician and survey by occupational health problem (N = 776).

	Intention to Disclose to Occupational Physician		Intention to Disclose to Survey	
Problems With:	Yes n (%)	No n (%)	Yes n (%)	No n (%)
Physical Health ^1^	708 (91.7%)	64 (8.3%)	727 (94.2%)	45 (5.8%)
Mental Health ^2^	665 (86.1%)	107 (13.9%)	703 (91.1%)	69 (9.0%)
Health-related factors affecting work	711 (92.3%)	59 (7.7%)	713 (92.4%)	59 (7.6%)
Use of medication ^3^	722 (93.6%)	49 (6.4%)	698 (90.4%)	74 (9.6%)
Lifestyle	701 (91.0%)	70 (9.1%)	700 (90.7%)	72 (9.3%)

Counts of NA—missing values are deleted from this table, superscript numbers (^1^, ^2^ and ^3^) indicate significance of differences (in proportion tests) between surveys and physicians in each category: ^1^
*p*-value = 0.074, ^2^
*p*-value = 0.003, ^3^
*p*-value = 0.024.

**Table 3 ijerph-18-01460-t003:** Ordinary least squares (OLS) regression results (with standard error) of intention to conceal from survey from occupational physician and intention to conceal in sum (N = 776).

	Dependent Variable		
	Intention to Conceal (0–10)	Intention to Conceal from Survey (0–5)	Intention to Conceal from Occupational Physician (0–5)
Gender (reference = Female)			
Male	0.369 *	0.330 **	0.018
	0.181	0.112	0.103
Other sex	−0.114	−0.495	0.358
	1.906	1.186	1.089
Age	−0.021 **	−0.011 **	−0.010 **
	0.007	0.004	0.004
Hospital (reference = Hospital A)			
Hospital B	0.061	0.084	0.002
	0.205	0.127	0.117
Hospital C	−0.028	0.056	−0.066
	0.189	0.117	0.108
Hospital D	0.510 *	0.06	0.452 **
	0.232	0.144	0.132
Education (reference = Higher education)			
No degree	1.033	0.65	0.388
	0.697	0.434	0.398
Primary education	0.839 *	0.254	0.577 *
	0.401	0.249	0.229
Secondary education	0.560 **	0.297 **	0.278 **
	0.182	0.113	0.104
Other education	0.497	0.456	0.033
	0.461	0.287	0.263
Score trust in physician ^1^	−0.080 **	−0.021 **	−0.058 **
	0.011	0.007	0.007
Relationship with Occupational Physician (OP) (reference = >5 years)			
Known OP for 3–4 years	−0.313	−0.135	−0.17
	0.202	0.125	0.115
Known OP for 1–2 years	−0.144	−0.051	−0.108
	0.207	0.129	0.118
Known OP less than 1 year	−0.189	−0.086	−0.122
	0.232	0.144	0.132
EQ5D VAS ^2^	0.002	−0.001	0.004
	0.005	0.003	0.003
General Mental Health (GHQ) ^3^	−0.038	−0.023	−0.012
	0.026	0.016	0.015
Has taken medication in last 2 weeks	0.332 **	0.137 *	0.194 **
	0.102	0.063	0.058
Medication dependence—used recreationally	0.387 *	0.164	0.215 *
	0.169	0.105	0.097
Alcohol Use Disorders (AUDIT-C ^4^ score)	−0.172 **	−0.066 *	−0.098 **
	0.044	0.027	0.025
Constant	4.462 **	1.655 **	2.736 **
	0.724	0.449	0.413

^1^ Adjusted Trust in Physician Scale, ^2^ Visual Analogue Scale (VAS), ^3^ General Health Questionnaire (GHQ), ^4^ Alcohol Use Disorders Identification Test (AUDIT-C). Note: * *p* < 0.05; ** *p* < 0.01.

## Data Availability

Due to the sensitive and confidential nature of the data, no data can be made available publicly.

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
