# Peer review of "Disclosure in Online vs. Face-to-Face Occupational Health Screenings: A Cross-Sectional Study in Belgian Hospital Employees"

_ijerph, 2021, doi:10.3390/ijerph18041460_

Round 1
Reviewer 1 Report
This is a very interesting paper, with relevant issues to be discussed for research and clinical practice in occupational health.
The abstract is adequate.
Introduction: It describes the state of art related to the subject and presents a relevant justification for the study. Objectives are clear.
Are there any references for the statements in the second paragraph?
Regarding the following sentence: "The employee has an incentive to exaggerate conditions in order to gain adjustments to his workload (e.g. feigning sleeping problems to avoid night work), or the opposite: underreport his alcohol misuse to ensure he can keep his job in the transport sector." Do the authors have any reference on that too? This afirmation is plausible and intuitive. However, is it a rule? Is there any epidemiological evidence on that? Sometimes, even when the worker has a severe or disabling condition, he/she can exaggerate a symptom due to fear, lack of trust, cultural or other social reasons, what can be misunderstood as "simulation", lie or dishonesty. However, doctor-patient relationship is complex and all these aspects are relevant for clinical practice. Trust is an essential aspect of doctor-patient relationship and a lie may not mean being dishonest, exactly.
Methods: "The survey contained two questions on whether the employee would honestly report a range of potential problems." Please specify exactly what questions were used to evaluate the main outcome (these questions are only partially written in table 2 later on, but it should be explicit in the methods). Also, the authors should provide more information on other tools and measures (parameters, cutoff scores, etc).
Criteria to include variables in regression models should also be explicit in the methods section.
Results: The meaning of scores 0-10 and 0-5 (table 2) is not clear for me. How were the scores calculated? It should also be described in the methods section.
Discussion: In my opinion, surveys will never be able to replace face-to-face occupational physicians' evaluation and in-person assessment of fitness for work. On the other hand, virtual surveys are really important for screening of occupational health exposures and diseases, for occupational health surveillance and to guide occupational health policies in organizations.
I believe a brief discussion about the meaning of the word "honesty" should be written in the discussion section. The term employees' "dishonesty" could be understood as a moral issue and it could suggest workers "misconduct". However, most of the "lies" said by workers/patients in consultations with doctors are due to fear, lack of trust, cultural characteristics or social reasons. A "lie" in a doctor-patient relationship must be understood in its complexity, considering several factors involved in it. Sometimes it is not even a conscious behaviour. The authors partially mention these aspects during the text, but it could be highlighted.
Surely, for the reader, it is clear that the questions used by authors to measure the outcomes actually included the specific term "honesty", and the outcome was evaluated by self-report. However, this measure might not be the best way to investigate the validity of workers information, either in surveys or face-to-face consultation. These aspects could be discussed as limitations.
Lastly, the discussion on occupational health online screenings should be seen as a complementary strategy, and not as a replacement for face-to-face consultation. If the authors agree with that, it should also be highlighted.
Reviewer 2 Report
IJERPH 1084735
Honesty in online vs. face-to-face occupational health screenings: a cross-sectional study in Belgian hospital employees
This study is clearly written and presents results that could be of practical usefulness for hospitals in conducting occupational health screenings. A few comments are below:
- Lines 30-33 briefly describe periodic health screenings but it would be helpful to have more information about these screenings since they are not done in every country. In Belgium, are ALL employees from all types of organizations (even outside of healthcare) required to undergo screenings? What happens with the information from the screening—how does the hospital use that information and/or does some other entity use that information? Are employee screenings anonymous or confidential? Does that vary from hospital to hospital? Can employees be fired based on their answers? What are the consequences of reporting truthfully if the answers reflect issues with the employee? Please provide more background to help us better understand employee motivation to lie or to be truthful. The background could be written in the introduction and/or in the discussion but I had a lot of questions about the screenings.
- Line 69 typo: “…of a questionnaire, it is not straightforward..”
- On lines 80 and 115 reference is made to “eligible” employees. Please describe the criteria for employees to be eligible for a yearly periodic health screening. And include what percentage of all hospital employees meet that eligibility for a yearly periodic health screening. In Table 1, it looks like 90% of respondents had a musculoskeletal complaint in the last 12m and 72% in the last 7 days. Was making an occupational injury complaint in the last 12 months part of the eligibility for participation in the study?
- Under Material and Methods, line 88, please add more detail about the two questions that were asked. Only in the title of Table 2 do we see the question text so please include the question text for the two questions in the Methods and indicate the response options (were responses yes/no?). How many total questions were respondents asked? Were the two questions about honesty the first two questions or where in the entire survey were those questions placed relative to the other questions?
- In the Methods please provide references/citations for each of the measures used and where they can be found for those who want to know more about them.
- In the Methods please indicate whether your survey was anonymous or confidential.
- Throughout the results and discussion, be careful to word the finds as hypothetical rather than using past tense. For example, on lines 129-133 it reads:
“…6% to 14% indicated to withhold information…”
“..198 employees (26%) stated being dishonest…”
“…143 employees (19%) stated being dishonest…”
However, since the questions were hypothetical, “Would you honestly point out a problem…” then the results must be worded to reflect the hypothetical nature of the question, rather than using past tense which implies that respondents are responding about their ACTUAL dishonest reporting. Therefore, reword as
“…6% to 14% indicated they would withhold information…”
“..198 employees (26%) reported they would be dishonest…” OR
“…143 employees (19%) indicated they would not honestly report…”
Same issue in the Discussion, line 166 should be reworded to, “…of the respondents indicated they would not always report honestly…” and in the Conclusions line 225, “…respondents indicated they would conceal information…” line 226, “..Respondents indicated they would be more inclined…”
- In line 227 please write out the scenario for “the opposite was true.” It’s not clear what the opposite is.
- In Table 3 please label the AUDIT-C as (alcohol use disorders) in the table rather than just as a footnote.
- In the discussion, you mention the possibility of false positives, where employees might report an issue to get accommodations. You could indicate that future studies could clarify the extent of reporting false positives vs. false negatives by asking more specific questions (for example, something like):
For false negatives:
“If you had a [physical health, mental health, medication use…] problem, would you report the problem [to your occupational physician/on an anonymous (or confidential) survey]? If not, why would you not report it?” You could list several reasons and include a write-in response option.
For false positives:
“If you did not have a [physical health, mental health, medication use…] problem, would you report [to your occupational physician/on an anonymous (or confidential) survey] that yes, you actually had a problem? If yes, why would you report yes even though you actually did not have a problem?” You could list several circumstances or reasons and include a write-in response option.
Or you could develop scenarios. It is likely that dishonest reporting has underlying reasons so it would add a lot to the research to better understand the reasons and circumstances under which employees would be dishonest in reporting, beyond their demographics and health status. For example, you mention the importance of anonymity or confidentiality or trust in the confidentiality of the information, fear of reporting due to consequences—so those you could measure the extent to which those are the reasons for reporting dishonestly.
- Since you captured staff positions, it would be interesting to examine whether certain staff positions say they would report dishonestly. Perhaps lower level staff fear for their jobs more than higher level staff, or some other categorization by staff position.
